# Representing Structural Isomer Effects in a Coarse-Grain Model of Poly(Ether Ketone Ketone)

**DOI:** 10.3390/polym17010117

**Published:** 2025-01-05

**Authors:** Chris D. Jones, Jenny W. Fothergill, Rainier Barrett, Lina N. Ghanbari, Nicholas R. Enos, Olivia McNair, Jeffrey Wiggins, Eric Jankowski

**Affiliations:** 1Micron School of Material Science and Engineering, Boise State University, Boise, ID 83725, USA; chrisjones4@u.boisestate.edu (C.D.J.); jennyfothergill@boisestate.edu (J.W.F.); rainierbarrett@boisestate.edu (R.B.); 2School of Polymer Science and Engineering, University of Southern Mississippi, Hattiesburg, MS 39406, USA; lina.ghanbari@usm.edu (L.N.G.); nicholas.enos@usm.edu (N.R.E.); olivia.mcnair@usm.edu (O.M.); jeffrey.wiggins@usm.edu (J.W.)

**Keywords:** thermoplastic, molecular dynamics, coarse-graining

## Abstract

Carbon-fiber composites with thermoplastic matrices offer many processing and performance benefits in aerospace applications, but the long relaxation times of polymers make it difficult to predict how the structure of the matrix depends on its chemistry and how it was processed. Coarse-grained models of polymers can enable access to these long-time dynamics, but can have limited applicability outside the systems and state points that they are validated against. Here we develop and validate a minimal coarse-grained model of the aerospace thermoplastic poly(etherketoneketone) (PEKK). We use multistate iterative Boltzmann inversion to learn potentials with transferability across thermodynamic states relevant to PEKK processing. We introduce tabulated EKK angle potentials to represent the ratio of terephthalic (T) and isophthalic (I) acid precursor amounts, and validate against rheological experiments: The glass transition temperature is independent to T/I, but chain relaxation and melting temperature is. In sum we demonstrate a simple, validated model of PEKK that offers 15× performance speedups over united atom representations that enables studying thermoplastic processing-structure-property-performance relationships.

## 1. Introduction

Thermoplastic composites are promising next-generation aerospace materials due to their low weight, high strength, and the ability to thermally weld parts for uniform, fastener-free joints [1,2,3,4,5]. These composites can be made from carbon fiber weaves or unidirectional tapes impregnated with thermoplastic, which is a polymer-based matrix material that softens when heated. Panels and parts are made from layering tapes into forms that undergo high-temperature consolidation. The resulting parts can be welded together by locally heating contacting surfaces at their interface [6].

Welding together small parts into larger structures presents both opportunities and challenges. Opportunity lies in the ability to mass produce the small parts, which can be individually checked for quality control. The challenge lies in ensuring the strength and failure behavior of the large structures, particularly at the welds, because no stress-bearing carbon fibers span these joints [6,7]. The strength of welds therefore depends on the entanglements of the polymer molecules that interpenetrate across the interface [8]. Understanding and improving thermoplastic composite welds, therefore, depends on a mechanistic understanding of how the matrix chemistry, polydispersity, defects, and surface preparation all influence the polymer structure that results from welding at a particular time, temperature, and pressure. With so many processing parameters to tune, exhaustively identifying the optimal combination is an intractably large design problem to tackle by trial and error.

Molecular dynamics (MD) simulations are a promising tool for serving as a surrogate to trial-and-error experiments and for providing mechanistic insight into thermoplastic welding. However, because the timescales of polymer dynamics are long (e.g., many minutes for some polymer crystallizations [9,10,11]), and because the length scales of structural periodicities can exceed many nanometers, MD simulations of atomic polymer representations are prohibitively expensive [12,13,14]. Computational costs for structurally complex, slow-relaxing systems compound because the time it takes to evaluate one time step forward in MD at best scales O(NlogN) with the number of simulation elements (atoms) *N*, and $N\approx 5\times 10^{4}–1\times 10^{6}$ atoms are usually needed to represent these polymers. Additionally, the more atoms in the system, the higher the number of time steps it takes for the system to relax to equilibrium. Consequently, a doubling of the length scales observed in a simulation requires 8× more simulation elements, which would in the best case require 8× more time steps to relax, and 16.6× more seconds per step, or a simulation that takes 133× as long to run.

Even worse are the time dynamics that matter for welding: in order for a strong weld to form, interpenetration of polymer chains (entanglement) needs to occur across the interface [15,16], as opposed to chemical bonding in epoxy thermoset matrices [17,18]. In order to accurately model polymer entanglements, and the role they play in thermoplastic welding and mechanical strength, two aspects that need to be considered are the entanglement length $\left(N_{e}\right)$ and the chain diffusion behavior in the presence of entanglements, both of which are described by the reptation model of Edwards and De Gennes [19,20]. Therefore, thermoplastic weld models need to include chain lengths long enough to result in the formation of entanglements and access reptation timescales. Here, we take a step towards this end by developing and evaluating a coarse-grained (CG) model that can better access time and length scales relevant to the thermal welding process of polymers than more detailed models [8,21].

There are multiple approaches for creating coarse-grained potentials, each with their own advantages and disadvantages, and each optimizing different aspects of the underlying physics and material properties. Several reviews discussing these coarse-graining methods exist; in particular, we direct the reader towards reviews from Gartner and Jayaraman [22] and Dhamankar and Webb [23], which enumerate coarse-graining approaches for polymers. We briefly describe three classes: force matching, relative entropy, and iterative Boltzmann inversion (IBI). Force matching aims to minimize the error between the forces in the coarse-grained model and the net forces in the atomistic system projected onto the coarse-grained mapping scheme, and therefore should result in an equivalent trajectory [24,25]. Relative entropy is an iterative method that minimizes statistical differences between the target and coarse-grained models [26,27]. While relative entropy can reproduce both equilibrium structure and forces from a target model, implementation is non-trivial, and the iterative updates to the coarse model are not as straight forward as either force matching or structure matching methods [23]. Boltzmann-inversion-based methods are designed to generate coarse-grained models that reproduce fine-grained structural distributions, such as the radial distribution function (RDF) and bond length, bond angle, and dihedral probability distributions, by iteratively learning intramolecular and intermolecular potentials [28]. IBI is commonly used for coarse-graining polymer systems [29,30,31,32,33]. A drawback of IBI is that a coarse forcefield derived at one state point is not generally transferable across other thermodynamic state points. However, it has been shown that structural distributions drawn from multiple state points can inform a transferable potential. This method, termed multi-state iterative Boltzmann inversion (MSIBI), developed by Moore et al., has been successfully applied to systems from relatively simple systems such as Lennard Jones fluids, propane, and water to more complex systems such as lipid bilayers [34,35,36,37,38].

### PEKK

We constrain our present modeling work to a case study of poly(ether-ketone-ketone) (PEKK), an aerospace composite matrix material that offers high mechanical strength, chemical resistance, and thermal stability [12,39,40]. It is a linear thermoplastic polymer in the polyaryl-ether-ketone (PAEK) family made from repeat units of phenyl rings with ether–ketone–ketone linkage groups as shown in Figure 1. The material properties of PEKK are dependent upon the connectivity of the polymer backbone, specifically the ratio between para and meta ketone linkages (Figure 1). For para linkages (Figure 1b), the $180^{\circ }$ relative orientation of ketone moieties results in straighter local chain segments. Meta linkages have a $120^{\circ }$ angle across their phenyl ring (Figure 1a) and reduce molecular symmetry, which influences crystallization kinetics and thermodynamic barriers to melting [11]. The relative amount of para and meta linkages is determined by the amounts of terephthalic (T) and isophthalic (I) precursors used in synthesis, respectively [11]. Consequently, PEKK can be thought of as a random co-polymer of para and meta isomers of ether–ketone–ketone repeat units, where the relative amount of each repeat unit is given by the T/I ratio. The “T/I ratio” we use throughout this work reports the relative amount of terephthalic linkages (e.g., a system with a T/I ratio of 80/20 is one with 80 percent para linkages and 20 percent meta linkages), and avoids nomenclature clashes with, for example, temperature.

Lower T/I ratios (i.e., decreasing the number of para linkages) results in lower $T_{m}$, slower crystallization kinetics, and decreased crystallinity [11,39,41,42]. For example, PEKK (T/I = 100/0) has a reported $T_{m}$ of 410 °C, while PEKK (T/I = 60/40) melts near 300 °C [39,41]. However, the T/I ratio has little effect on the glass transition temperature ($T_{g}$) [41]. As a consequence of this T/I ratio effect, the melt temperatures and crystallization kinetics of PEKK can be tuned by choosing the best suited T/I ratio for the application. In the case of high-throughput manufacturing, lower T/I ratios might be preferred to induce faster processing times via the decrease in $T_{m}$. Additionally, PEKK materials with lower processing temperatures help to avoid the undesirable effects of melt-state degradation that have been reported in PAEK polymers [43,44,45]. Alternatively, if a larger degree of crystallinity is favored, then high T/I ratios may be preferred. Regardless of the application, it is apparent that the T/I ratio of the PEKK matrix material offers another adjustable parameter in an already extensive design space of thermoplastic composite manufacturing.

In this work, we present a CG model of PEKK, derive a CG forcefield using MSIBI, and introduce a simple way to model T/I ratio effects. We show that this model matches the structural predictions of more detailed models but with 2×–15× lower computational cost. The open-source, freely available tools developed here lay the foundation for studying weld structures in PEKK-based composites.

## 2. Models

In this section, we describe the united atom (UA) model used to train the coarse-grain (CG) model and the unique representation challenges in coarse-graining para- and meta-oriented PEKK monomers.

### 2.1. United Atom PEKK

We use a UA model to generate “target” structural distributions of PEKK across thermodynamic state space, which is used to inform the CG model. UA models represent all atoms in a molecule explicitly as spherical simulation elements, except for non-alcohol hydrogens, whose representations of mass and charge are “united” into the heavy elements to which they are bonded [46]. Non-bonded interactions are modeled with 12-6 Lennard-Jones potentials [47] truncated at σ=2.5. We employ a General Amber Forcefield (GAFF) [48] parameterization of the bond, angle, dihedral, and non-bonded interactions. The details of these parameterizations are available at https://github.com/rsdefever/GAFF-foyer/tree/master/gafffoyer/xml, which is the specific force field file imported by the present simulation workflows (https://github.com/chrisjonesBSU/pekk-cg-model).

### 2.2. Coarse-Grained PEKK

We represent each repeat unit of PEKK with three spherical simulation elements. Each element represents a phenyl ring plus its connecting group (“E” for ether and “K” for ketone, Figure 2). The resulting chain topology is a linear sequence of repeating E−K−K elements. The resulting forcefield is relatively simple, requiring parameterization of only two bond constraints, two angles, two dihedrals, and three pair-wise non-bonded interactions (Table 1).

Using this CG representation, the atomistic details of the para and meta ketone moieties are abstracted away (Figure 2), which raises the question of how to best capture the effect of the T/I ratio on PEKK polymer properties in the CG model. Keeping track of separate para and meta ketone simulation elements is one possible approach, but one which would complicate the otherwise simple forcefield. The non-bonded interactions of these chemically identical coarse simulation elements should be similar, which raises questions about how to constrain training with MSIBI: do we enforce identical pairwise interactions between these simulation elements, or do we allow MSIBI to find unique potentials while introducing challenges with how we interpret them? That is, because the differences in the para and meta linkages manifest as differences in bonded constraints between coarse simulation elements, we aim to only increase model complexity when it addresses relevant physics. To help answer these model design questions, we measure which specific components of the CG model depend on the T/I ratio and create a coarse-graining strategy that accurately accounts for these effects only where necessary. The current body of work around CG models of PEKK is relatively small. To the best of the author’s knowledge, the only other reported CG model of PEKK is presented by Chattaraj and Basu, which uses a mapping scheme slightly different from this work, and does not attempt to model the T/I ratio within the CG forcefield [49]. For both the UA and CG models, we report distances in units of simulation element diameters (σ) and energy (ϵ) in units of the strongest non-bonded interaction of the underlying UA model, where σ=0.33996 nm and ϵ=0.87864 kJ/mol. Unless indicated otherwise, all temperatures are reported in reduced units, where $T=\frac{kT}{\epsilon }$.

## 3. Materials and Methods

We apply one high-level method (MSIBI [34]), through which we implement MD simulations of UA PEKK, MD simulations of CG PEKK, and structural analysis. The resulting CG model is further validated by evaluations of $T_{g}$ and stress relaxation times—both experimentally and computationally—as a function of the T/I ratio. A summary of the mathematical expressions used in this work is available in Appendix A.

### 3.1. Molecular Dynamics Simulations

Molecular dynamics simulations of both UA and CG representations are carried out using the HOOMD-Blue [50] simulation engine and signac [51] data space management framework. Simulations are performed on the Fry high-performance computing cluster at Boise State University, using NVIDIA P100 and V100 GPUs. All the simulations reported here, both UA and CG, are performed in the NVT ensemble (constant number of particles *N*, constant volume *V*, and constant temperature *T*), using the MTK velocity Verlet implementation of Nosé–Hoover chains [52,53] with a step size of dt=0.0003. We generate initial topologies using the mBuild Python package [54]. We parameterize UA systems using the foyer Python package [55] and the General Amber Forcefield (GAFF) [48] and subsequently unite hydrogen atoms into their bonded neighbors, summing their masses and partial charges into these simulation elements. No other modifications to forcefield parameters are performed. Partial charges are generated using antechamber with the AM1-BCC charge model [56] and integrated using particle–particle particle–mesh (PPPM) Ewald summation [57]. Simulations are performed until equilibration as measured by autocorrelation analysis of the potential energy time series. Scripts and details of models and methods are available at https://github.com/chrisjonesBSU/pekk-cg-model.

### 3.2. Learning Coarse-Grained Potentials with MSIBI

In this work, target bond lengths and bond dihedrals yield simple distributions suitable for fitting to established functional forms, and, as a result, we simply take the Boltzmann inverse of these distributions following
(1)$$V\left(x\right)=-k_{b}T\ln\left[P(x)\right],$$
where P(x) is the target distribution, $k_{b}$ is Boltzmann’s constant, *T* is the temperature, and V(x) is the potential of mean force. For bond lengths, V(x) is fitted to a harmonic potential
(2)$$V_{bond}\left(l\right)=\frac{1}{2}k{\left(l-l_{0}\right)}^{2},$$
where *k* is the spring constant, *l* is the bond length, and $l_{0}$ is the equilibrium bond length. Because the dihedral distributions show only one broad peak, V(x) is fitted to a harmonic dihedral
(3)$$V_{dihedral}\left(\phi \right)=\frac{1}{2}k\left(1+d\cos\left(\phi -\phi_{0}\right)\right),$$
where *k* is the spring constant, *d* is the sign factor, ϕ is the dihedral angle, and $\phi_{0}$ is the equilibrium dihedral angle. The distributions for bond angles do not yield simple distributions suitable for fitting to known functional forms, and the flexibility affordable by tabulated potentials is needed. Therefore, bond angles and non-bonded pair distributions are optimized following a series of CG simulations performed iteratively at the same thermodynamic state points as UA simulations, and the resulting structural distributions compared against the target distribution. After each iteration, the CG potentials are updated according to
(4)$$V_{i+1}\left(x\right)=V_{i}\left(x\right)-\frac{1}{N}\sum_{s}\alpha_{s}k_{B}T_{s}\ln\left[\frac{P_{s}^{i}\left(x\right)}{P_{s}^{*}\left(x\right)}\right],$$
where $V_{i}$ is the CG potential at iteration *i*, *N* is the number of state points used, $\alpha_{s}$ is a state point-specific weighting factor, $k_{B}$ is the Boltzmann constant, $T_{s}$ is the state point temperature, and $P_{s}\left(x\right)$ is the structural distribution corresponding to the potential V(x). $P_{s}^{*}\left(x\right)$ denotes the target distribution. In this case, the parameter $\alpha_{s}$ serves two purposes. First, it acts as a damping factor, limiting the magnitude of the potential updates at each iteration. Second, it is a state point weighting factor where each state can be assigned a separate value of $\alpha_{s}$, which controls how much that state point influences the final potential. Iterations are run until $P_{s}\left(x\right)$ approaches $P_{s}^{*}\left(x\right)$, at which point V(x) converges to its final form.

In this work, the angle potentials are derived using a single state, with α=0.7 over 15 IBI iterations. The intermolecular pair potentials are obtained with all four states weighted equally, using α=1.0 over 20 iterations. We follow a sequential forcefield process, optimizing one forcefield component first, which is fixed during optimization of the next component. Structure-matching methods such as IBI and MSIBI typically optimize one interaction type at a time in the order of their relative strength, which follows as [58,59]:
$$V_{stretching}\to V_{bending}\to V_{pair}\to V_{dihedral}$$

#### Measuring Model Success

Quantitative measurement of the match between the target UA and CG bond, angle, dihedral, and pair-wise distributions is reported using $f_{fit}$, which is calculated by
(5)$$f_{fit}=1-\frac{\sum_{x_{start}}^{x_{cut}}\left(|P^{i}\left(x\right)-P^{*}\left(x\right)|\right)}{\sum_{x_{start}}^{x_{cut}}\left(|P^{i}\left(x\right)|+|P^{*}\left(x\right)|\right)},$$
where P(x) is the distribution resulting from the CG model and $P^{*}\left(x\right)$ is the target distribution [34]. In the case of a perfect match, the numerator goes to zero, and $f_{fit}=1.0$. To quantify CG validation of chain statistics, we compare the average squared radius of gyration $\left(\langle R_{g}^{2}\rangle \right)$, squared end-to-end distance $\left(\langle R_{e}^{2}\rangle \right)$, and persistence length $\left(\langle \ell_{p}\rangle \right)$ between the CG and UA models. For each T/I ratio of 100/0, 80/20, 70/30, and 60/40, single-chain simulations of both the coarse-grained model and target model are run for 20 repeat units over a temperature range of *T* = 4.0 to 6.0, with increments of 0.5. The persistence length measurements are performed using the mdanalysis package [60,61]. Scripts described here are available at https://github.com/chrisjonesBSU/pekk-cg-model.

### 3.3. Validation

The CG model of PEKK is validated against the equilibrium structural distributions of UA PEKK, as described in “Section Measuring Model Success” above, the degree to which $T_{g}$ is independent of the T/I ratio, and the degree to which stress relaxation times depend on the T/I ratio. In order to determine the $T_{g}$ of the CG model, we use the mean-square displacement (MSD) to obtain diffusion coefficients $D=\frac{\partial MSD}{6\;\partial t}$. For each T/I ratio, CG simulations of 50 polymers with 20 repeat units, corresponding to a molecular weight of 6002 Da, are run over a temperature range of T= 0.2 to 4.0 in increments of 0.2, and their MSD measured using the freud analysis package [62]. For each T/I ratio, D is plotted against *T* and $T_{g}$ is identified as the temperature where D begins to diverge from D=0, which is indicative of the onset of cooperative polymer mobility. We measure the $T_{g}$ in the present work with the above diffusivity criterion instead of specific-volume-based methods [63,64,65,66,67] because of the relatively low equilibrium specific volume precision observed in NPT CG simulations [68,69]. The method is further justified by Henry et al., who demonstrate that determining $T_{g}$ from diffusion coefficients in CG models works and recreates experimental measurements in thermoset polymers [70].

To validate the CG model’s representation of how the T/I ratio affects $T_{m}$, we simulate polymer stress relaxation and compare against experimental stress relaxation rheology. This indirect comparison is chosen due to the difficulties in measuring $T_{m}$ due to the finite size effects of the solid–liquid interface on melting [13]. Practically, because $T_{m}$, crystallization kinetics, and relaxation times all depend on the same underlying mechanism of chain stiffness determining cooperative movement length scales [71,72,73], we focus here on relaxation dynamics to infer the degree to which a coarse model can capture the T/I ratio effect on all three of these properties. Stress relaxation experiments as a function of the T/I ratio are performed experimentally and computationally, and both methods are explained below.

Rheological stress relaxation experiments were performed on an ARES G2 parallel plate rheometer (TA Instruments, New Castle, DE, USA) with a forced air convection oven and five thermocouples to monitor the sample temperature. PEKK copolymers purchased from Arkema with T/I ratios of 60/40, 70/30, and 80/20 (Mw of 65 k, MWD of 2.5, pellets. Kepstan^®^ 6002, 7002, and 8002, respectively) were characterized. Stainless steel plates with a diameter of 8 mm separated by a 1.0 mm were used. Samples were loaded at 360 °C to avoid thermo-oxidative degradation. All experiments had a 1 mm gap and samples were pre-sheared to allow for homogeneity, good contact with the plates, and to remove potential air bubbles. The samples were then heated to 380 °C for 3 minutes to erase any order remaining in the melt, and then cooled to 360 °C. An instantaneous strain of 200.0% was applied to the samples and stress was monitored over time, collecting data at 5000 points per second to observe the rapid relaxation. The time at which 200.0% strain was reached was taken to be t=0. The degree to which the stress relaxed was measured and normalized by the initial stress value at t=0.

To qualitatively simulate these experiments, and show relaxation time trends in the CG model, for each T/I ratio we initialize systems of 200 polymers with 20 repeat units each, corresponding to a molecular weight of 6002 Da, at a density of $1.38\;\frac{g}{{\mathrm{cm}}^{3}}$ where the chains begin in an elongated and aligned state. Each system is held at the same temperature of 2.0 T while the chains relax from the initial non-equilibrium state to an equilibrium state. To show chain relaxation time trends, we measure the decay of an order parameter rather than measuring stress directly. We use the nematic order parameter S2 as a measure of chain backbone linearity and the decay of the nematic order over time as a qualitative analog to stress relaxation decay from experimental results. S2 is a description of orientational ordering, and is often used as a measurement of polymer chain alignment [74,75], and experimental work has shown a relationship between nematic order and equilibration relaxation times [72,73]. The freud Python library is used to calculate the nematic order parameter [62].

In summary, the success of the CG model across the T/I ratio range is measured structurally and dynamically. First, we determine if the bonded and non-bonded structural distributions are accurately re-created. Second, we ensure key polymer chain statistical metrics are reproduced by the coarse-grained model, including $\langle R_{g}^{2}\rangle$, $\langle R_{e}^{2}\rangle$ and $\langle \ell_{p}\rangle$. Third, we validate the T/I ratio-dependent angle potentials by comparing chain relaxation against rheological measurements and testing for $T_{g}$ invariance.

## 4. Results and Discussion

In this section, we report and discuss the structural distribution functions generated by UA simulations of PEKK, followed by structural and dynamic validation of the CG model and its performance. We show that only the E−K−K angle potentials vary with the T/I ratio and describe the potentials derived by MSIBI. With the CG model, we show reproduction of bulk morphology across the training state points, as well as good agreement with individual chain statistics. We find that the chain relaxation times of the CG model as a function of the T/I ratio qualitatively match experimental stress relaxation measurements. Finally, we find raw performance improvements of 2×–15× over UA models.

### 4.1. United Atom Simulations

Single-chain UA simulations are run to obtain the target bond length, angle, and dihedral distributions used for generating the CG bonded potentials. Chain lengths of 20 repeat units are run at a low density of $0.0003\;\frac{g}{{\mathrm{cm}}^{3}}$ (i.e., vacuum) at a temperature of 415 °C. The single-chain simulations are run for 7.3 ns with a time step of 0.145 fs. In order to observe the effect that the T/I ratio has on the bonded interactions, simulations are performed for each T/I ratio of 100/0, 80/20, 70/30, and 60/40. The sequence of T and I monomers are assigned randomly down the length of the polymer chain, weighted by the target T/I ratio. In order to ensure effective sampling, five different random sequences are generated for each T/I ratio with single-chain simulations run separately for each sequence. We measure the bond length, angle, and dihedral distributions for each T/I ratio by averaging over the replicate single-chain trajectories.

Bulk systems of 50 PEKK oligomers consisting of six repeat units using a T/I ratio of 100/0 are run to obtain the target radial distribution functions needed for generating the CG pair potentials. Only a single T/I ratio is used, as we expect any significant differences in pair-wise distributions as a function of the T/I ratio to ultimately be the result of chain structures arising from intra-chain interactions. In other words, the presence of meta linkages does not create differences in the chemical makeup or mapping of our coarse-grained beads in a way such that their underlying pair interactions are effectively different. Polymers below typical molecular weight values for PEKK are used in order to increase computational efficiency in reaching equilibrated target trajectories. Previous work shows that coarse-grained pair potentials derived from oligomers are transferable to systems of larger molecular weights [76]. The state points used are described in Table 2. State point A is a unique case where the system consists of two chains of 16 repeat units each in a vacuum. Inclusion of this state is discussed in Appendix A, where we report the impact on the CG model’s ability to match the radius of gyration and end-to-end distance. State points B, C, and D include two states with a temperature half-way between $T_{g}$ and $T_{m}$ at reported values for both amorphous and crystalline densities, and a state above melting at an amorphous density [77].

The bond, angle, and dihedral distributions as a function of the T/I ratio obtained from the target single-chain trajectories are shown in Figure 3. The bond length and dihedral distributions appear relatively insensitive to the T/I ratio and we elect not to have T/I dependencies on these model components for simplicity, though it is possible that adding such dependencies back in may improve model quality. The E−K bond distributions are nearly identical over the entire T/I ratio range. We measure a 4% variance in the K−K bond length across the T/I ratios sampled here. Given our experience with coarse models, we do not expect this to significantly affect the bulk structure or properties of the coarse-grained model [78]. Additionally, the K−E−K angle distribution shows no notable dependence on the T/I ratio. Significant differences arise in the E−K−K angle distribution, which clearly exhibits the most sensitivity to the T/I ratio, with an emergence of a second peak near 1.3 radians once the T/I ratio drops below 100/0. The intensity of this second, smaller angle peak grows as the T/I ratio decreases, while the peak at 2.2 radians decreases. The K−E−K−K dihedral appears to vary with the T/I ratio; however, the dihedral prefactor only varies from 10 to 13 energy units across this T/I range. Due to these very weak interactions and relatively small variance, we choose to omit this T/I dependency for simplicity and consistency in the present model.

### 4.2. Coarse-Grained Model

#### Coarse-Grained Potentials

Here, we summarize the complete CG model, where Figure 4 shows the components of the CG model that utilize table potentials and Table 3 lists the parameters used in fitting functional forms. Together, these make up the complete CG model as defined in Table 1.

Figure 4 highlights the angle and pair potentials resulting from IBI and MSIBI, respectively. Figure 4a shows four different E−K−K angle potentials in the CG model as a function of the T/I ratio. An emergence of a second, meta-stable energy well at smaller angles is observed as the T/I ratio decreases. Additionally, the energy barrier between the large-angle and small-angle wells appears to slightly decrease with decreasing T/I ratios. For the K−E−K angle, we only derive a single potential that is used across all T/I ratios studied in this work. Potentials derived using IBI and MSIBI may not necessarily conform to commonly used functional forms, and can often have multiple wells and peaks that may be unphysical. Here, we find sensible potentials that are consistent with a soft-core Lennard-Jones-like pair potential. The E−E and K−K pair potentials exhibit a similar shape, with the E-E beads having a slightly smaller effective radius and a deeper energy well. In the E−K pair, we observe a similar effective radius to K−K, with a significantly shallower energy well. The parameters obtained from fitting the Boltzmann inversion (Equation (Equation 1)) of bond length (Equation (Equation 2)) and dihedral (Equation (Equation 3)) distributions are shown in Table 3.

### 4.3. Coarse-Grained Model Validation

Figure 5 illustrates the pair-wise RDF comparison between the CG and UA models for each pair across all four states.

Good agreement between the distributions is observed at all four state points. In Appendix A, we provide detailed snapshots of each plot shown in Figure 5, as well as a summary of the $f_{fit}$ scores obtained using Equation (Equation 5).

Figure 6 shows the performance in matching the single-chain angle distributions over the T/I ratio range.

The CG model accurately reproduces the emergence of smaller angle peaks with decreasing large angle peaks for the E−K−K angle while achieving $f_{fit}$ values of 0.98 or better at each T/I ratio. The K−E−K distributions across the T/I ratio range are also accurately recreated. These observations verify that the tabulated CG E−K−K angle potential is sufficient for recreating the T/I ratio-dependent angle distributions observed in the UA model.

#### Single-Chain Structure

Figure 7 reports the measured results for persistence length ($l_{P}$), squared end-to-end distance ($R_{e}^{2}$), and squared radius of gyration ($R_{g}^{2}$) for a single T/I ratio of 80/20 across a reduced temperature span of T= 4 to 6.

As temperature is increased, we observe monotonically increasing trends for all three metrics in the UA model. Measurements of $l_{P}$, $R_{e}^{2}$, and $R_{g}^{2}$ in the CG model generally agree well in magnitude and variance with the UA model, but we do observe some discrepancies. In the CG model, $l_{P}$ is systematically 10% higher for T≤5, and then no longer monotonically increases for T>5. For $R_{e}^{2}$, we also do not observe the same monotonically increasing trend in the UA model, nor the significantly lower $R_{e}^{2}$ for T=4. While the UA model does not demonstrate strong end-to-end distance increases with temperature, the qualitatively different temperature dependence of the CG model is worth noting, and could be an indicator of some underlying physics that are abstracted away in the CG model. The radius of gyration $R_{g}^{2}$ measurements of the CG model match better than the other two metrics: we observe close agreement in both magnitude and monotonic increase with *T*. The slope of the $R_{g}^{2}$ increase is not as high in the CG model as the UA model.

Figure 8 shows the same measurements at a constant reduced temperature of T= 5.0 over the T/I ratio range studied. Across these three metrics, only the 60/40 T/I ratio is distinct, giving a lower $l_{P}$ measure than the higher ratios in the UA model. The CG model matches these trends: The 60/40 ratio gives the lowest $l_{P}$, and is otherwise constant with increasing T/I ratio at this temperature. The CG model is systematically about 10% higher in $l_{P}$ at this temperature, though this offset is not particularly concerning as it is less than a tenth of the monomer diameter. Across $R_{e}^{2}$ and $R_{g}^{2}$, we observe close agreement in both magnitude and variance between the CG and UA models. A more detailed statistical analysis of these measures is provided in the Appendix A. Unfortunately, experimental data for these properties are not readily available, so we are unable to identify if the lack of a trend of these properties with respect to the T/I ratio is to be expected, or an indication that the model is missing some of the underlying physics that govern PEKK polymer conformations. Overall, we find the general agreement of the single-chain statistics in the CG model—which are emergent properties that the CG model was not biased towards—with the UA model to be encouraging. It is true that the CG model does not show identical monotonic increases in $l_{P}$, $R_{e}^{2}$, and $R_{g}^{2}$ to the UA model, and an investigation of whether such agreement can be trained for in MSIBI, or whether these discrepancies influence properties, is left for subsequent work.

### 4.4. Modeling the Effect of T/I Ratios

Figure 9 shows the self-diffusion coefficient (D) over a range of temperatures of the CG model for each T/I ratio. Where the D transitions from zero to non-zero is indicative of the onset of $T_{g}$ [70,79]. We can see that the transition point shows no significant changes based on the T/I ratio used, which is consistent with the observed behavior of $T_{g}$ in PEKK being invariant to the T/I ratio.

In Figure 10, we show a comparison between chain relaxation times as a function of the T/I ratio between the CG computational model and experimental results obtained from rheological stress relaxation. While these two experiments are measuring different properties, they are both ultimately measuring the same relaxation process from non-equilibrium states to relaxed states. Both measurements, S2 order parameter decay in the computational model and stress decay in the experiments, clearly show a dependence on the T/I ratio, where polymer melts with larger T/I ratios relax more slowly than those with lower T/I ratios. A T/I ratio of 100/0 is not available experimentally; however, the computational CG model with this ratio maintains the trend of increasing relaxation time as the para linkage backbone content increases. While the computational model cannot access the same timescales as experiments, this consistency between experiment and computational models is encouraging, and indicates that the T/I ratio effect is captured by the E−K−K angle potential (Figure 4a).

### 4.5. Computational Performance

Raw performance was measured by time steps per second (TPS) using NVIDIA Tesla-P100 GPUs with 12 GB of VRAM, and the comparison between UA and CG models is shown in Figure 11. TPS improvement ranges from a factor of 2 at small system sizes to a factor of nearly 15 at system sizes of 30,000 monomers, where the CG model has a TPS of 851 while the UA model is only 54 TPS.

This performance improvement is an underestimate of overall performance improvement because the smoother free-energy landscape of the coarse model will facilitate faster relaxation times than a UA model of the same number of repeat units. At present, we lack a prescriptive time-mapping between coarse and detailed models and leave such quantification to future work. However, the observed raw performance improvement measured here, plus the expectation of improved relaxation dynamics, informs our claim that this validated model offers significant sampling advantages over UA models.

## 5. Conclusions

We develop a simplified CG model that accurately represents the molecular structure and bulk morphology of PEKK. This model is trained on a relatively small sample set: four T/I ratios of single chains and four bulk trajectories. With only one T/I ratio-dependent component—the E−K−K angle-potential—this model captures the effect of the T/I ratio on chain relaxations, while not influencing $T_{g}$, in agreement with experiments. This approach demonstrates that a modest set of training data is sufficient for informing simplified CG models of thermoplastics that capture complex chemical dependencies (here, T/I ratio), and future work can examine functional interpolations across T/I space rather than individual tabulated potentials for each ratio. The factor of 15× speedup at larger system sizes is promising for investigating PEKK structure in experimentally relevant volumes. However, slight differences in the temperature dependence of $l_{P}$, $R_{e}^{2}$, and $R_{g}^{2}$ in the CG model indicate that there could be room for improving the CG model with additional training. Future work incorporating new training measures into MSIBI to accomplish this offers promise to both improve CG models and incorporate recent advances in machine learning into coarse-graining. This model is extensible, where obtaining new potentials for $U_{EKK}\left(\theta ,T/I\right)$ would only require running a single-chain UA simulation followed by a short IBI optimization of only one interaction—both of which are computationally inexpensive tasks. The approach developed here has future application for capturing structural isomer effects in CG models of other polymers containing phenylene backbones, such as poly(phenylensulfide), poly(phenylenediamine), and other polymers in the polyaryl-ether-ketone family of thermoplastics. Fuller exploration of interface dynamics during fusion bonding and evaluation of entanglement dynamics with our CG model will be the subject of future work.

## Figures and Tables

**Figure 1 polymers-17-00117-f001:**
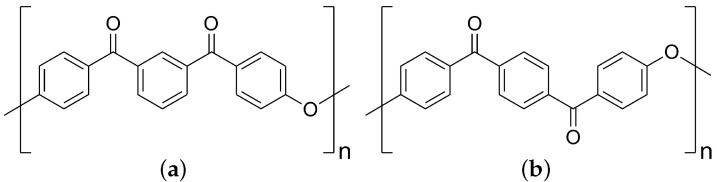
Two isomers of the PEKK monomer. (**a**) Monomer containing meta linkages synthesized from isophthalic acid (I). (**b**) Monomer containing para linkages synthesized from terephthalic acid (T). The higher the T/I ratio of monomers used to synthesize the copolymer, the straighter the chains and the higher the $T_{m}$.

**Figure 2 polymers-17-00117-f002:**
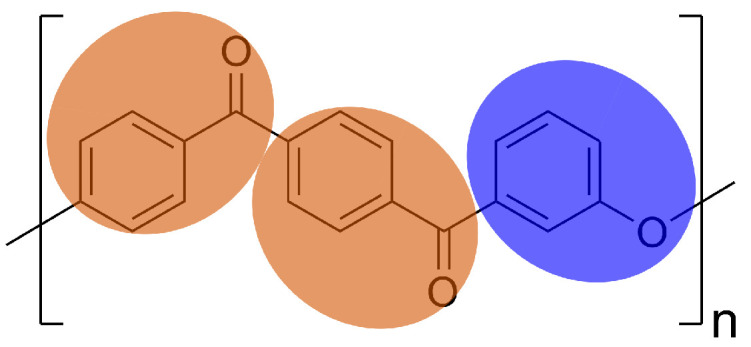
Coarse-grained mapping scheme with E (blue) and K (orange) beads. The mapping scheme chosen loses information about the orientation of consecutive ketone groups in the monomer.

**Figure 3 polymers-17-00117-f003:**
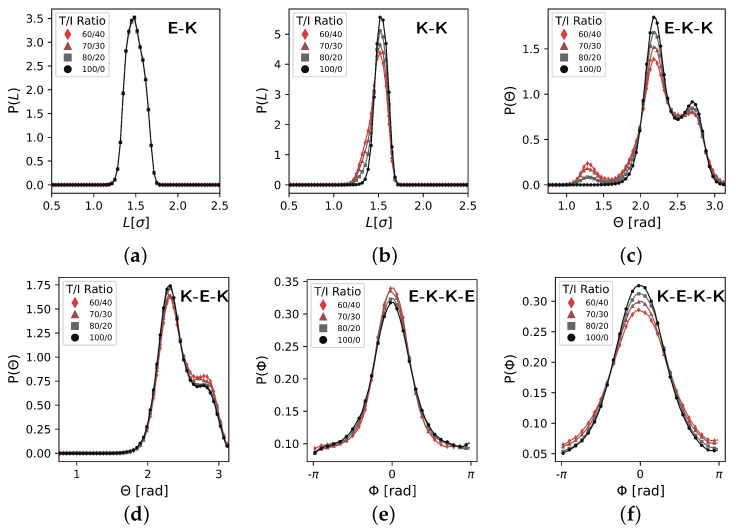
Across all T/I ratios, the only intra-chain distribution that changes shape is the EKK angle potential. We see slight variations in width for the K−K bond and K−E−K−K dihedral distributions. Intra-chain distributions are obtained from united atom single-chain simulations at a reduced temperature of T=6.5. (**a**) E−K bonds, (**b**) K−K bonds, (**c**) E−K−K angles, (**d**) K−E−K angles, (**e**) E−K−K−E dihedrals, and (**f**) K−E−K−K dihedrals.

**Figure 4 polymers-17-00117-f004:**
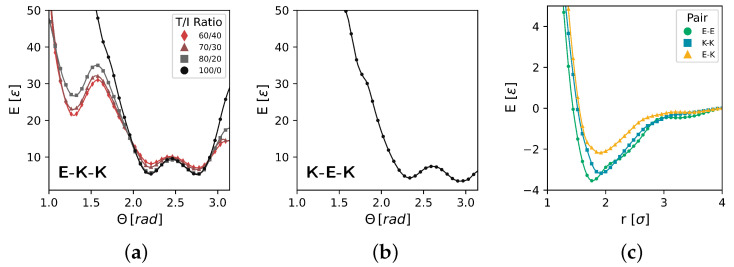
Table potentials obtained from MSIBI for the CG PEKK model. (**a**) Four separate CG E−K−K angle potentials are shown as a function of the T/I ratio. As the T/I ratio decreases, we observe the formation of a small-angle energy well. (**b**) The CG K−E−K angle potential is shown, where we derive only a single tabular form, as opposed to a T/I ratio-dependent set of potentials. (**c**) Non-bonded pair potentials are shown for all three bead-type interactions.

**Figure 5 polymers-17-00117-f005:**
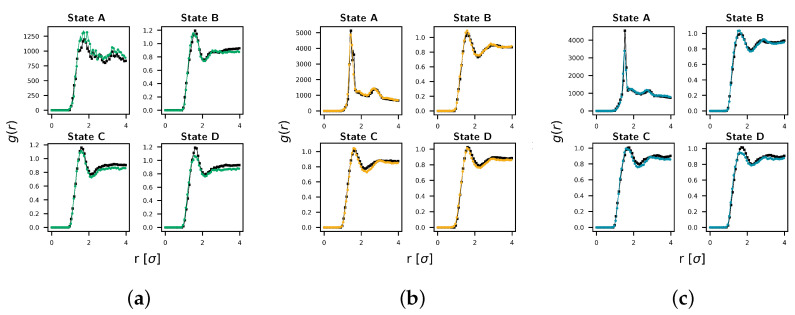
Across all pairs and state points, we observed good agreement between radial distribution functions. Target RDFs from the UA model are shown in black, and RDFs from the CG model are shown by the colored lines. Pair correlations between bead types are as follows: (**a**) E−E, (**b**) E−K, and (**c**) K−K.

**Figure 6 polymers-17-00117-f006:**
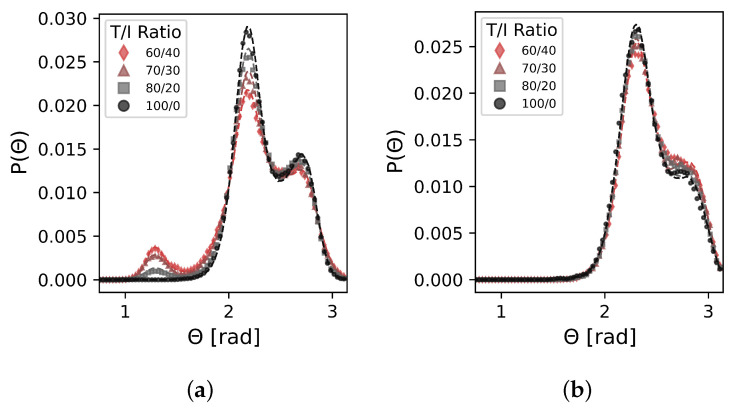
Comparison of (**a**) target UA E−K−K and (**b**) K−E−K angle distributions (dashed lines) and CG model angle distributions (symbols). We see good agreement for both across all T/I ratios studied. K−E−K distributions are matched at all T/I ratios even without using a T/I-dependent K−E−K angle potential.

**Figure 7 polymers-17-00117-f007:**
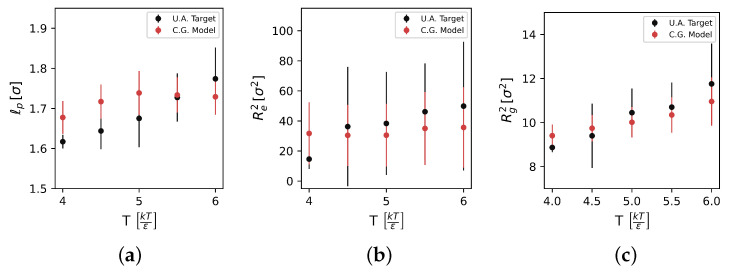
(**a**) Persistence length, (**b**) squared end-to-end distance, and (**c**) squared radius of gyration comparisons between UA (black) and CG (red) models over a temperature range with a T/I ratio of 80/20.

**Figure 8 polymers-17-00117-f008:**
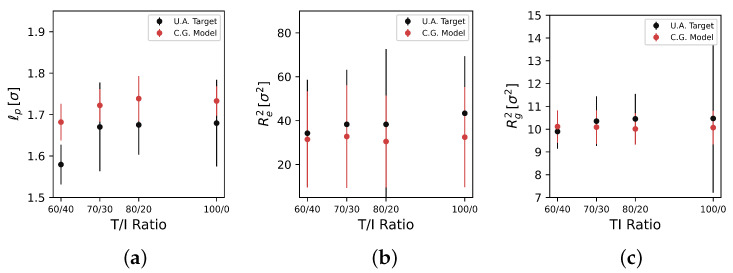
(**a**) Persistence length, (**b**) squared end-to-end distance, and (**c**) squared radius of gyration comparisons between UA (black) and CG (red) models over the T/I ratio range studied at a constant reduced temperature of $T=5.0\frac{kT}{\epsilon }$.

**Figure 9 polymers-17-00117-f009:**
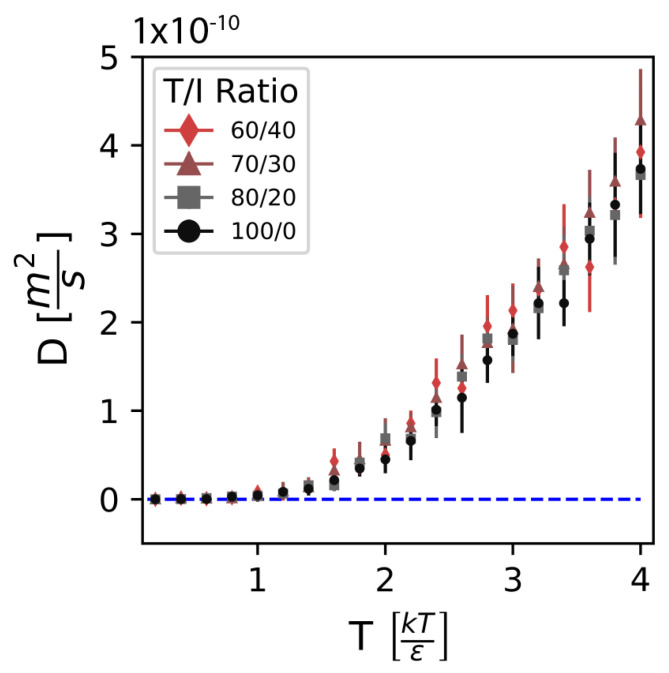
Self-diffusion coefficients measured across several temperatures for each T/I ratio. $T_{g}$, identified where D begins to exponentially increase from D=0 (dashed blue line), shows no notable dependence on T/I ratio.

**Figure 10 polymers-17-00117-f010:**
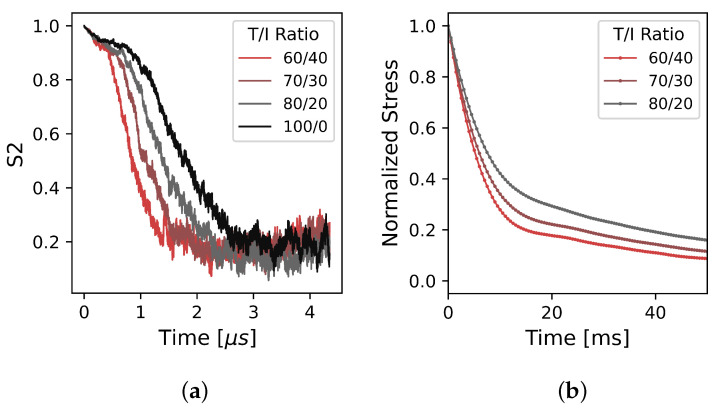
Qualitative comparison of chain relaxation times as a function of T/I ratio with (**a**) nematic order parameter decay of the CG model and (**b**) normalized stress decay from experimental results.

**Figure 11 polymers-17-00117-f011:**
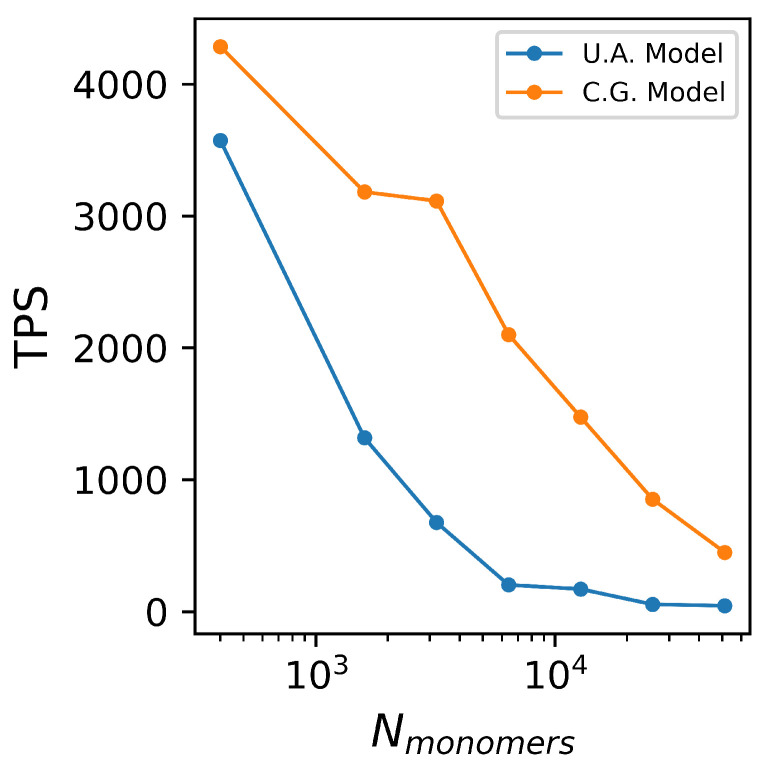
Time steps per second comparison between the united atom (UA) and coarse-grained (CG) models run on NVIDIA Tesla-P100 GPUs.

**Table 1 polymers-17-00117-t001:** Coarse-grained model summary.

Interaction	Group
Bonds	E-K, K-K
Angles	E−K−K, K−E−K
Dihedrals	E−K−K-E, K−E−K−K
Pairs	E-E, K-K, E-K

**Table 2 polymers-17-00117-t002:** United atom simulations are performed at four thermodynamic states spanning two temperatures and three densities.

State	Temperature °C	Density $\frac{g}{{\mathrm{cm}}^{3}}$
A	414	0.0003
B	255	1.27
C	255	1.35
D	414	1.27

**Table 3 polymers-17-00117-t003:** Parameters for CG potentials for bond stretching (V(l)) and dihedral distributions (V(ϕ)).

CG Potential	Parameters
$V_{EK}\left(l\right)$	$k=850\epsilon ,l_{0}=1.47\sigma$
$V_{KK}\left(l\right)$	$k=1450\epsilon ,l_{0}=1.53\sigma$
$V_{EKKE}\left(\phi \right)$	$k=16\epsilon ,\phi_{0}=0,d=-1$
$V_{KEKK}\left(\phi \right)$	$k=13\epsilon ,\phi_{0}=0,d=-1$

## Data Availability

Data and scripts for simulation, visualizations, and analysis are available at https://github.com/chrisjonesBSU/pekk-cg-model/.

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
