# Peer review of "Representing Structural Isomer Effects in a Coarse-Grain Model of Poly(Ether Ketone Ketone)"

_polymers, 2025, doi:10.3390/polym17010117_

Round 1
Reviewer 1 Report
Comments and Suggestions for Authors The author(s) have studied the structural isomer effects in a coarse grain model of Poly(ether-ketone-ketone) (PEKK) that has potential use in aerospace industries. The paper is an interesting topic for polymers. However, the paper needs some revision before its acceptance. My comments are as follows: 1. Abstract was completely ambiguous. Coherency is missing in the abstract. Rewrite it. 2. The quality of all figures should be improved. 3. The author(s) used many mathematical expressions. Hence, a separate nomenclature section could be included in the manuscript or in the supplementary. 4. The motivation of the study, research gap, and objectives should be discussed in detail. 5. The testing methods were not fully described. It seems none of the testing was complying with the standard testing methods. 6. The discussions and conclusions should be separated. Results and discussion is to be improved a lot. 7. The limitations of the present work, future scope may be discussed in detail in the conclusion section. 8. Check the title. It may be REPRESENTING STRUCTURAL ISOMER EFFECTS IN A COARSE-GRAIN MODEL OF POLY(ETHER KETONE KETONE). Comments on the Quality of English LanguageThe English could be improved to more clearly express the research.
Reviewer 2 Report
Comments and Suggestions for Authors
The authors developed a coarse-grained (CG) model of poly(etherketoneketone) (PEKK) based on the multi-state iterative Boltzmann inversion method. Their CG model can accurately replicate the bulk morphology and polymer chain structure of an underlying united-atom model, and captures key T/I-dependent effects, including the observed trends in stress relaxation and the invariance of the glass transition temperature. This work is well done, valuable and could be published after considering some critical remarks:
1. The authors claimed that the CG model they developed can achieve the computational efficiency necessary to explore polymer diffusion and entanglement in welded interfaces. However, as the CG molecular dynamics simulations they performed are only for the systems with relatively short chain lengths (20 repeat units). Its validity for entangled systems needs to be verified.
2. The authors claimed that “we do not consider the slight variance in the K-K bond length distribution width large enough to significantly affect the bulk structure or properties of the coarse-grained model”. Why? Are there any evidences?
3. From Figure 3(f), I find that the effect of T/I ratio on K-E-K-K dihedrals is also rather significant. However, its impact on CG model is not discussed.
4. In Figure 4a, which result is for IBI?
5. In Figure 4a, I only see a single curve of KEK angle potential. How the authors obtained the conclusion of “The KEK angle potential is independent of T/I ratio”.
6. In Page 9, the author stated that “The E-E and E-K pair potentials exhibit similar shape with the E-E beads having a slightly smaller effective radius and deeper energy well.” Why the “K-K” pair is not mentioned?
7. In Table 3, why CG potentials of angles and pairs are not included?
8. In Figure 7 and 8, Some discrepancies are observed for UA and CG models. However, it is not clearly explained.
